# Neuronal Bmal1 regulates retinal angiogenesis and neovascularization in mice

Vijay K. Jidigam[1], Onkar B. Sawant[1,2], Rebecca D. Fuller[1], Kenya Wilcots[1,3], Rupesh Singh[1], Richard A. Lang [4,5] & Sujata Rao [1,6✉]

Circadian clocks in the mammalian retina regulate a diverse range of retinal functions that allow the retina to adapt to the light-dark cycle. Emerging evidence suggests a link between the circadian clock and retinopathies though the causality has not been established. Here we report that clock genes are expressed in the mouse embryonic retina, and the embryonic retina requires light cues to maintain robust circadian expression of the core clock gene, Bmal1. Deletion of Bmal1 and Per2 from the retinal neurons results in retinal angiogenic defects similar to when animals are maintained under constant light conditions. Using two different models to assess pathological neovascularization, we show that neuronal Bmal1 deletion reduces neovascularization with reduced vascular leakage, suggesting that a dysregulated circadian clock primarily drives neovascularization. Chromatin immunoprecipitation sequencing analysis suggests that semaphorin signaling is the dominant pathway regulated by Bmal1. Our data indicate that therapeutic silencing of the retinal clock could be a common approach for the treatment of certain retinopathies like diabetic retinopathy and retinopathy of prematurity.

[1] Department of Ophthalmic Research, Cole Eye Institute, Cleveland Clinic, Cleveland, OH 44195, USA. [2] Eversight, Cleveland, OH 44103, USA. [3] Department of Chemistry, Cleveland State University, Cleveland, OH 44115, USA. [4] Division of Developmental Biology, Cincinnati Children's Hospital, Cincinnati, OH, USA. [5] Department of Ophthalmology, College of Medicine, University of Cincinnati, Cincinnati, OH, USA. [6] Department of Ophthalmology, Cleveland Clinic Lerner College of Medicine of Case Western Reserve University, Cleveland, OH 44195, USA. ✉email: RAOS7@ccf.org

All living organisms have an internal time-keeping mechanism that controls behavioral and physiological responses in anticipation of daily changes in the environment. In mammals, a central pacemaker clock within the suprachiasmatic nucleus (SCN) coordinates alignment between external time-giving cues such as the day–night cycle with multiple peripheral oscillators throughout the body. These oscillators or peripheral clocks regulate gene transcription to alter tissue function and thus remain in phase with the central oscillator. At the molecular level, this is achieved through the activity of the core clock genes that maintain autoregulatory feedback loops in which the oscillating outputs regulate their expression. The product of clock genes Bmal1 and clock (or its paralog NPAS2) drive the expression of the negative regulator *Per1-3* (Period 1–3) and *Cry1-2* (Cryptochrome 1–2). The CRY/PER heterodimers inhibit CLOCK/BMAL1 transcriptional activity and expression. Fine-tuning and reinforcement of this core loop are ensured by additional mechanisms and, importantly, by posttranslational events, such as phosphorylation/dephosphorylation, which regulate nucleocytoplasmic shuttling and degradation of clock proteins. This self-sustained process has an approximate period of 24 h (circa) and, by driving gene expression, sets the rhythms of cellular activity. How this translates into well-synchronized functions of cells within tissues or organs remains to be investigated.

The mammalian retina also contains an autonomous circadian system that functions independently of the SCN[1–5]. The retinal clock regulates various aspects of retinal physiology critical for processing of visual information and thus optimizing retinal activity with the diurnal changes in light intensity. However, it is not clear when the retinal circadian system is functional[6,7], though some impacts of clock function on retinal development has been described[8–10]. The retinal clock can be entrained by light both in vitro and in vivo, suggesting that light plays a critical role in the regulation of the retinal clock.

The developing retina is photoreceptive before any visual photo-transduction can occur due to the expression of several light-sensitive opsins. We and others have shown that these opsins regulate critical developmental pathways required for visual function[11,12]. The retina is also highly vascularized with an inner and outer retinal vasculature to support the high metabolic demands of neuronal activity. Retinal angiogenesis is dependent on the retinal ganglion cells and in mice lacking RGCs, the retina remain avascular[13]. However, it is not known if this phenotype is due to the loss of molecular cues that are derived from the activity of the retinal ganglion cells or due to loss of a scaffolding substrate that is required for the growth of the vessels. In this study we investigated whether a retinal neuronal clock can drive developmental and pathological angiogenesis by genetically deleting Bmal1 and Per2 from the developing retina. Our study establishes a role for the neuronal light driven clock in regulation of retinal angiogenesis and further supports the idea that the neuronal clock but not the endothelial clock is an important mediator of pathological angiogenesis. Thus, these studies suggest that the clock genes are a good therapeutic targets for the treatment of diseases like retinopathy of prematurity and other proliferative retinopathies.

## Results

### Rhythmic expression of circadian clock genes in the embryonic mouse retina is light dependent. 
To assess whether core clock genes are expressed in the fetal retina, transcript levels of clock genes Bmal1, Per2, Clock, and Cry1 were measured over 24 h starting at embryonic day E16 (E16) (Fig. 1a–f). We performed a cosinor fit model analysis to determine whether there was any significant 24-h variation in the expression of these clock genes

(Fig. 1a–e). The transcript levels for Bmal1 ($p = 0.0030$) (Fig. 1a) and Cry1 ($p = 0.0348$) (Fig. 1d) oscillate robustly over 24 h. To assess if the transcriptional oscillation persists under constant conditions, we placed the pregnant dams in constant darkness (DD) (12D:12D) at E13 and harvested the embryonic retina at E16. Under these conditions, the oscillations of Bmal1 and Cry1 transcripts in the embryonic retina was completely abolished (Fig. 1a–d). The expression of Per2 and clock transcript was similar in the LD and DD group. The effect of DD on rhythmicity as well as other parameters like mesor, amplitude, and acrophase was examined and compared to LD conditions. Bmal1 ($p = 0.019$) transcripts show a significant reduction in amplitude, while Cry1 ($p = 0.056$) though reduced, is not significant (Fig. 1e). Furthermore, the acrophase for the Cry1 and Clock transcript is shifted with a delay in Bmal1 and Per2. To validate the transcriptional data, retinal lysates from the embryo were probed with a BMAL1 antibody. BMAL1 protein is detected in the retina at E16; however, due to the low amount of overall protein, it was difficult to assess whether the protein levels oscillate over 24 h (Fig. 1f). Bmal1 transcript remains significantly lower in the retina even at postnatal day. Interestingly, whether the animals are kept in constant darkness (from E13-E16) or constant light (from E15-P4.5), Bmal1 expression is always reduced compared to the LD control (Fig. 1g, h). Thus, our data suggest that in the developing retina, Bmal1 expression is dependent on the environmental light.

### Disruption of Bmal1 and Per2 expression in the retinal neurons results in retinal angiogenesis defects. 
Retinal neurons can partly influence the growth of the retinal vasculature through a light-mediated signaling pathway[11]. Neurons in the retina are critical for the proper formation of the retinal vessels under physiological and pathological conditions[14–18]. Retinal neurons can partly influence the growth of the retinal vasculature through a light-mediated signaling pathway[11,16–18]. To determine whether the oscillations in Bmal1 levels have any physiological relevance, we assessed whether retinal angiogenesis is affected by disrupting Bmal1 expression. We used two types of manipulations to change Bmal1 levels, one was changing the light environment and the other was through genetic deletion of Bmal1. Animals were maintained in constant light (LL) or under standard mouse room lighting conditions (LD). At P7.5, typically in the control animals, the superficial vascular plexus grows and extends from the optic nerve head (ONH) and reaches the periphery (Fig. 2k). Though the vessel growth appears to be normal in the LL animals (Fig. 2m), there is a significant increase in the retinal vascular density (Fig. 2l, n). These findings further validate that environmental light significantly alters retinal angiogenesis[11].

Next, we wanted to test the hypothesis that light-mediated signals regulate retinal angiogenesis by modulating the circadian clock in the neurons. Bmal1 is a key regulator of the circadian clock genes, thus deletion of Bmal1 abrogates clock function. Additionally, Per2 is a negative regulator of the Bmal1 dependent clock gene expression, therefore we sought to delete both Bmal1 and Per2 from the retinal neurons expecting to see opposing effects on retinal angiogenesis. We used the Chx10Cre transgenic animals in which the Cre is expressed in the retinal progenitor cells (RPCs) at E13[19]. In these animals, the Cre expression is mosaic, thus sometimes resulting in partial deletions. At P7.5, vessel density in the superficial layer is significantly ($p = 0.033$) higher in the Bmal1$^{FL/FL}$; Chx10Cre animals compared to their Bmal1$^{FL/FL}$ cohorts (Fig. 2a–e). The increased vessel density is similar to what was observed when animals were maintained under LL conditions (Fig. 2o).

At P21, when the retina is completely vascularized, the hyperproliferative phenotype of the superficial layer remains

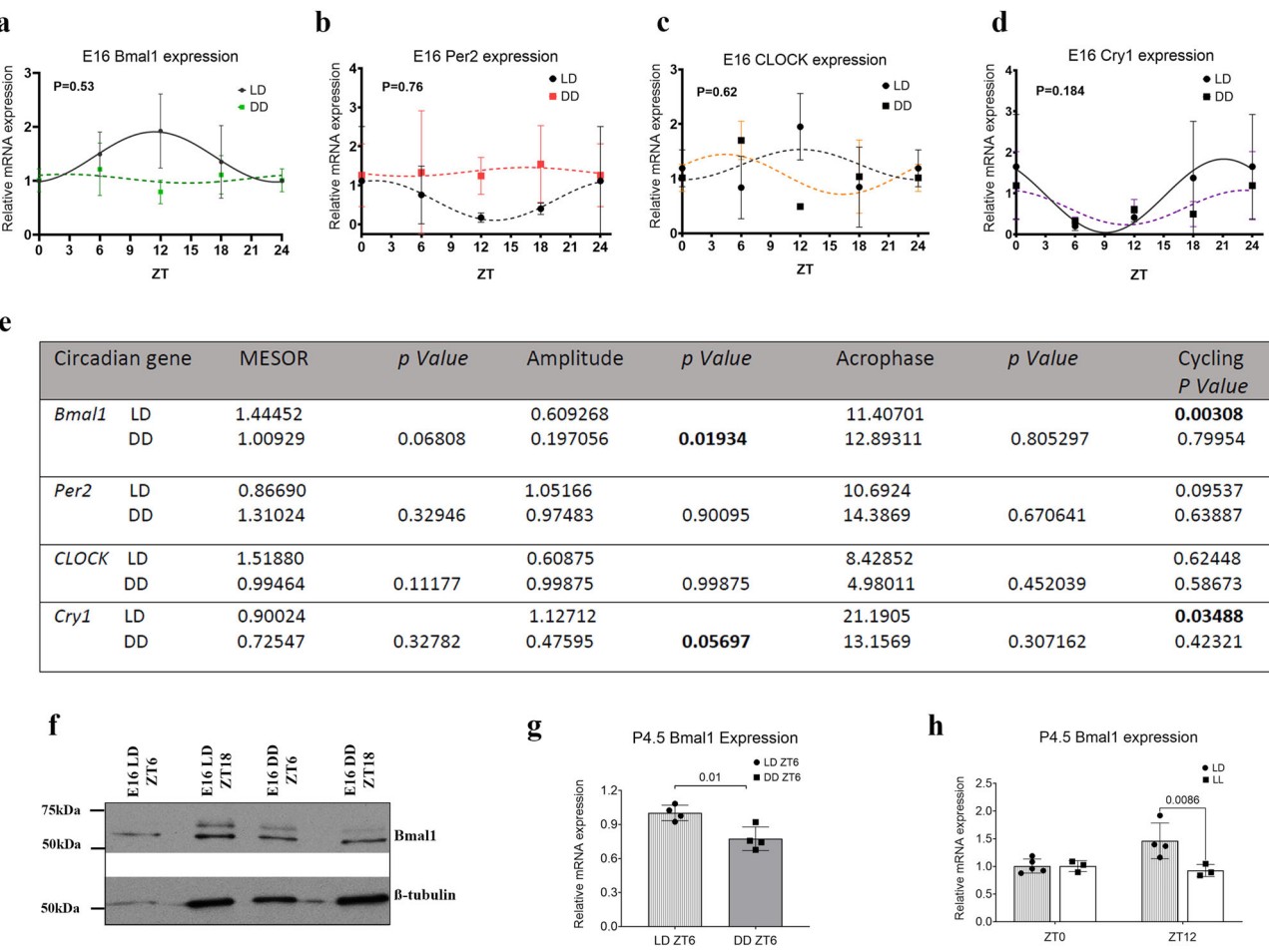

**Fig. 1 Clock gene expression and rhythmicity is light dependent.** Pregnant dams were housed under LD (12 L:12D) or DD (12D:12D) from E13 to E16. **a–d** Relative mRNA expression of *Bmal1* (**a**), *Per2* (**b**), *Clock* (**c**), and *Cry1* (**d**) in embryonic retina from LD and DD animals. *p*-values indicate group differences by two-way ANOVA (**a**, **b**, **c**, **d**). Cosinor analysis was used by best curve fitting of the gene expression data to check rhythmicity as shown in the solid line (**a–d**). Both Bmal1 (**a**) and Cry1 (**d**) show significant oscillation under 12L:12D conditions. **e** Table indicates the individual mesor, amplitude, and acrophase values for the different transcripts between the LD and DD conditions. Significant *p*-values are highlighted in bold. Bmal1 and Cry1 rhythmicity is lost under DD conditions. Each time point represents the mean with SEM from five embryos. **f** Western blots of E16 retinal lysates probed with Bmal1 and β-tubulin (housekeeping protein) under LD and DD conditions at ZT6 and ZT18. Note there is an overall reduction in Bmal1 protein. **g**, **h** Bmal1 transcript is significantly reduced at P4.5 when animals are maintained under constant light or dark conditions. Error bars represent SEM.

persistent, but the intermediate (inner plexiform layer) and deep layer vessels (outer plexiform layer) do not exhibit any defects (Fig. 2a'–i'). These results suggest that neuronal Bmal1 is required for maintaining the superficial vascular plexus. In contrast, loss of Per2, a negative regulator of Bmal1, results in reduced vessel density at P7.5 (*p* = 0.039) (Fig. 2f–j). However, this difference in vessel density normalizes by P21 and the superficial layer vasculature from the Per2^FL/FL; Chx10Cre animals are indistinguishable from wild animals (Fig. 2j'–2q'). Interestingly, at P21 the deep layer vessels are significantly affected, suggesting that the neuronal Per2 regulates the growth of the deeper layer vessels (Fig. 2o', r'). These results indicate that both Bmal1, the positive regulator of the circadian clock and Per2, the negative regulator exhibit opposing effects on murine retinal vascular development and influence each vascular plexus differently.

**Neuronal clock genes plays a crucial role in neovascularization in OIR.** Dysregulated crosstalk between the neurons and the vasculature is a significant factor contributing to proliferative vascular pathologies[20]. We used the murine model of oxygen-induced retinopathy to investigate whether dysregulation of the neuronal clock can result in vascular pathologies. In this model,

exposure of animals to hyperoxia from P7.5 to P12.5, induces profound capillary regression in the central retina. At P12.5, animals are returned to normal air resulting in hypoxic conditions that drive neovascularization[21].

In the, animals, hyperoxia causes increased vessel regression at P12.5. Despite the increased capillary regression, the rate of vessel regrowth is faster in the mutants (Fig. 3o). By P17.5, the retinas in the Bmal1^FL/FL; Chx10Cre animals are completely vascularized (Fig. 3c, d, h), unlike the control animals (Fig. 3a, b, h). By P20.5, there are still avascular regions in the control animals, while in the mutants, the retinal vasculature appears relatively normal (Fig. 3o) (Supplementary Fig. 1). Importantly, in the Bmal1^FL/FL; Chx10Cre animals, there is an overall reduction in pathological neovascularization at P17.5 (Fig. 3h). Low body weight can have an impact on the severity of the phenotypes observed in Oxygen Induced Retinopathy (OIR) animals, however, there was no difference between the Bmal1^FL/FL and Bmal1^FL/FL; Chx10Cre animals (Supplementary Fig. 3). Similarly, in the Per2^FL/FL; Chx10Cre animals there is an overall reduction in the avascular area (Fig. 3j–n). Thus, despite the increased regression of the capillary, the recovery is faster in the absence of neuronal Bmal1 or Per2. This data supports the idea that local dysregulation of the

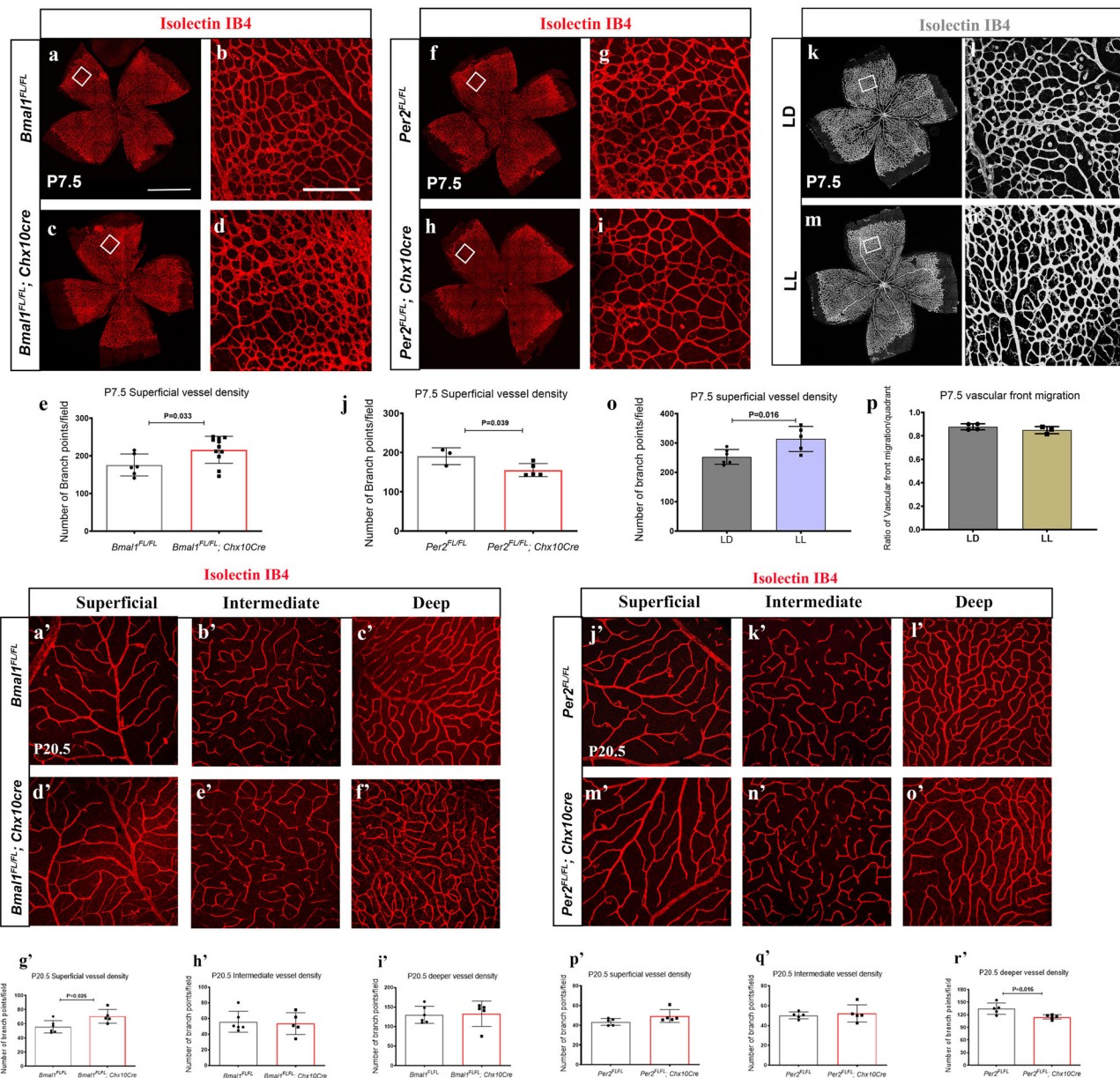

**Fig. 2 Bmal1 and Per2 have opposing effects on developmental angiogenesis. a, c, f, h, k, m** Retinal flat mounts labeled for endothelial cells with isolectin IB4. **b, d, g, i, l, n** represent higher magnification images of boxed regions to illustrate differences in vessel density at P7.5. Bmal1$^{FL/FL}$; Chx10Cre animals show a significant increase in vessel density in the remodeling area (b, d, e), whereas Per2$^{FL/FL}$; Chx10Cre animals show a significant decrease in vessel density (g, i, j) when compared to Bmal1$^{FL/FL}$ and Per2$^{FL/FL}$ controls, respectively. **k–p** Pups raised in constant light from ~E15-P7.5 exhibit significant vessel hypertrophy compared to the controls (l, n, o). Vessel migration is unaffected in the retina of the LL animals (p). **a'–o'** Representative isolectin stained images of Bmal1$^{FL/FL}$ (a', b', c'), Bmal1$^{FL/FL}$; Chx10Cre (d', e' f'), Per2$^{FL/FL}$ (j', k', l') and Per2$^{FL/FL}$; Chx10Cre (m', n,' o') from superficial (a', d', j', m'), Intermediate (b', e', k', n'), and deep layer (c', f', l', o'). (g', h', i', p', q', r') quantification of vessel density in Bmal1 (g', h', i') and Per2 mutants (p', q', r'). Note that only the superficial layer is affected in the Bmal1$^{FL/FL}$; Chx10Cre animals at P20.5 while only the deeper layer is significantly altered in the Per2$^{FL/FL}$; Chx10Cre animals. Significance was determined using a two-tailed *t*-test. Error bars represent SEM. *n* = 3–11. The scale bar on flat mounts and field images are 800 μm (**a**) and 50 μm (**b**), respectively.

clock protects the retina from further damage and appears to limit the damage caused by hyperoxia.

To further confirm that the neuronal clock is truly responsible for driving pathological neovascularization, we deleted Bmal1 from the endothelial cells using the tamoxifen inducible endothelial-specific Pdgf-icreER transgene (Bmal1; ECKO). Bmal1 deletion (tamoxifen injected from day of birth to P2.5) in the endothelial cells was confirmed using a Cre mediated fluorescent reporter (tdTomato) and by performing digital droplet PCR on the sorted endothelial cells (Supplementary

Fig. 2). In the Bmal1; ECKO animals, there is no difference in the capillary regression or the neovascularization compared to the littermate animals that were treated similarly (Fig. 3e, f, i). Thus, these results suggest an important role of neuronal clock genes in regulating the pathological overgrowth of the vessels in proliferative retinopathies.

**Bmal1 regulates neovascularization in adults**. Bmal1 appears to play a role in stimulating the growth of aberrant vasculature, as

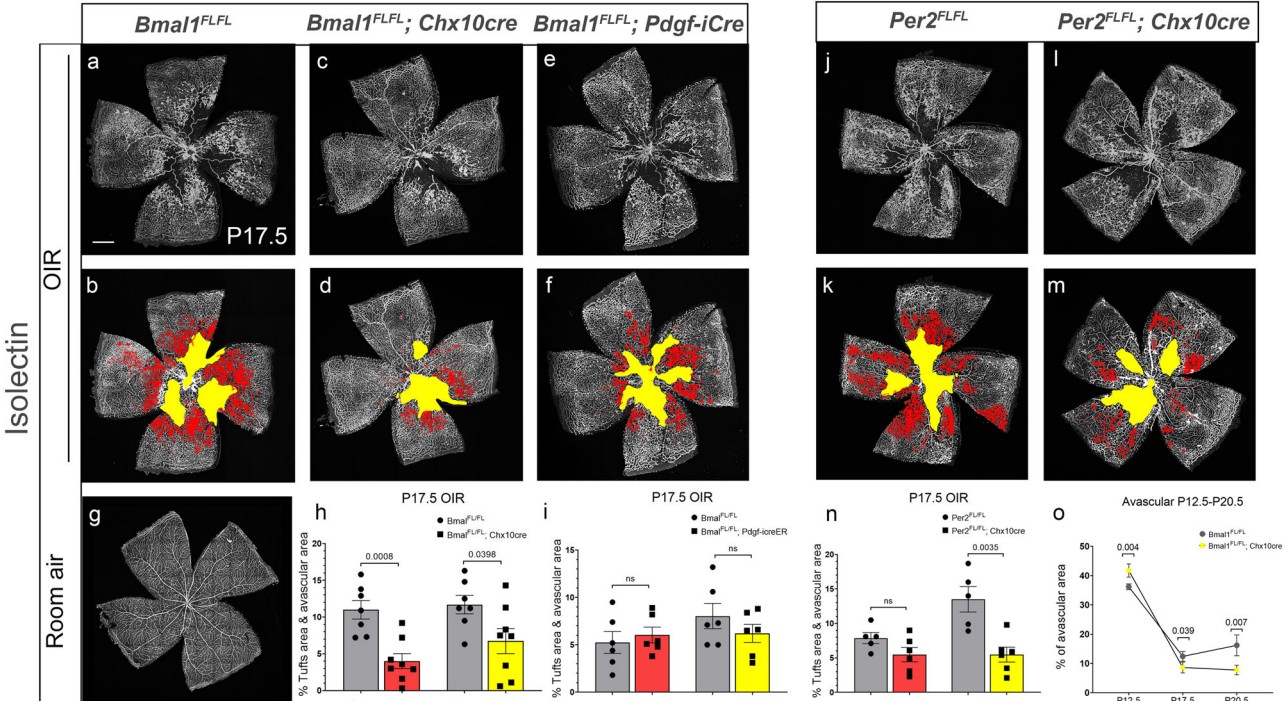

**Fig. 3 Neuronal clock drives pathological neovascularization.** Pups were exposed to hyperoxia from postnatal day (P) 7.5 to P12.5, moved to room air, and assessed at P17.5. **a–d** Retinal flat mounts from Bmal$^{FL/FL}$ (**a, b**) and Bmal1$^{FL/FL}$; Chx10Cre (**c, d**) labeled with isolectin. **b, d, f, k, m** Retinal images indicate areas of vaso-obliteration (VO) outlined in yellow and neovascular tufts in red. **g** Representative image of retinal flat-mount from room air control animals. **h** The VO and NV were automatically quantified using deep-learning segmentation software. Both VO and NV tufts in the Bmal1$^{FL/FL}$; Chx10Cre mutant animals (**d, h**) is significantly reduced when compared to Bmal1$^{FL/FL}$ controls (**b, h**), a similar reduction of VO area was observed in Per2$^{FL/FL}$; Chx10Cre (**l, m, n**) when compared to controls (**j, k, n**). In contrast, Bmal1 deletion from endothelial cells from day of birth did not have any effect on VO and NV (**e, f, i**), compared to control (**a, b, i**). **o** Quantitation of the rate of revascularization in the Bmal1$^{FL/FL}$ (gray circle) versus the Bmal1$^{FL/FL}$; Chx10Cre animals (yellow circle). Hyperoxia results in a significant increase in VO in the Bmal1$^{FL/FL}$; Chx10Cre mutants compared to the controls (gray circle), but the rate of vessel regrowth in the mutants is faster (yellow circle). *p*-values were acquired using two-tailed *t*-tests. Error Bars represent SEM. *n* = 5–8. Scale bar = 500 μm.

seen in the OIR animals. In the OIR model, retinal neovascularization occurs in the inner retina, and it is possible that the neuronal Bmal1 is essential during development but has no function in adults. To test whether neuronal Bmal1 plays a role in adult retinal neovascularization, we used laser to induce choroidal neovascularization (CNV) in 6-month-old mice. In laser-induced CNV animals, the regrowth of the vessels occurs from the choroidal endothelial cells[22]. CNV growth was assessed using fluorescein angiography at day 7, 14, and 21 post injury. The hyperfluorescent areas in FA images (Fig. 4b–d) is significantly reduced in the Bmal1$^{FL/FL}$; Chx10Cre animals at all ages examined (Fig. 4f–h). Furthermore, quantitation of these FA images shows a significant reduction in FA leakage in the Bmal1$^{FL/FL}$; Chx10Cre animals compared to the controls, suggesting that the CNV lesions are less leaky (Fig. 4i). In addition, we used optical coherence tomography (OCT) imaging to measure the volume of the CNV lesions in vivo, over the 3 weeks. We used multiple b-scan images for each lesion to calculate the lesion volume. Similar to the FA measurements, by day 7, 14, and 21 post injury, the lesion volume is significantly reduced in the Bmal1$^{FL/FL}$; Chx10Cre animals (Fig. 4j–r). This data confirms that Bmal1 mediated signaling is involved in neovascularization during CNV.

**Distinct patterns of Bmal1 occupancy in neural retina at P3 vs P5.** To gain insights into the neuronal genomic targets of Bmal1 we performed a chromatin immunoprecipitation analysis at P3 and P5. Though the retina is not entirely avascular at P3, it is predominantly neuronal. We reasoned that by comparing Bmal1

genome occupancy between these ages we will gain insights into the function of Bmal1 in the neural retina[23] before vascularization and post-vascularization. At P3 we identified 2367 possible Bmal1 candidate binding sites, which drastically drops to 241 by P5 (Fig. 5b). The distributions of the peaks departed from the actual distribution of functional domains, indicating that peaks were not randomly located on the genome but are enriched in specific functional domains. To validate whether these peaks corresponded to known Bmal1-binding sites, we probed the P3 and P5 neural retina core datasets for enriched motifs using Cistrome and Memechip tools. We used the Bmal1-binding site motif CACGTG with flexible C/T and G/T sequence specificity at position 3 and 4, respectively (Fig. 5a). At P3, only 85 Bmal1 targets (Supplementary Table 2) representing ~3.6% of the called peaks in the neural retina had Bmal1-binding motifs and at P5 this number dropped to 24 targets (Supplementary Table 3). Ingenuity Pathway analysis and panther analysis tool revealed that Bmal1 target genes at P3 are enriched in pathways like semaphorin signaling, Cadherin signaling, Rho family GTPases, calcium transport signaling pathways (Fig. 5c). In addition, Bmal1 regulates the expression of genes involved in cytoskeleton signaling, angiotensin signaling and synaptic tissue communication signaling at P3, while at P5 TGFß signaling related pathway are shown to be regulated. Additionally, Bmal1 is bound to the genomic regions containing key transport regulators.

**Evolutionary conservation as an indicator of relevance for Bmal1-bound regions.** The ChIP-sequencing data shows that

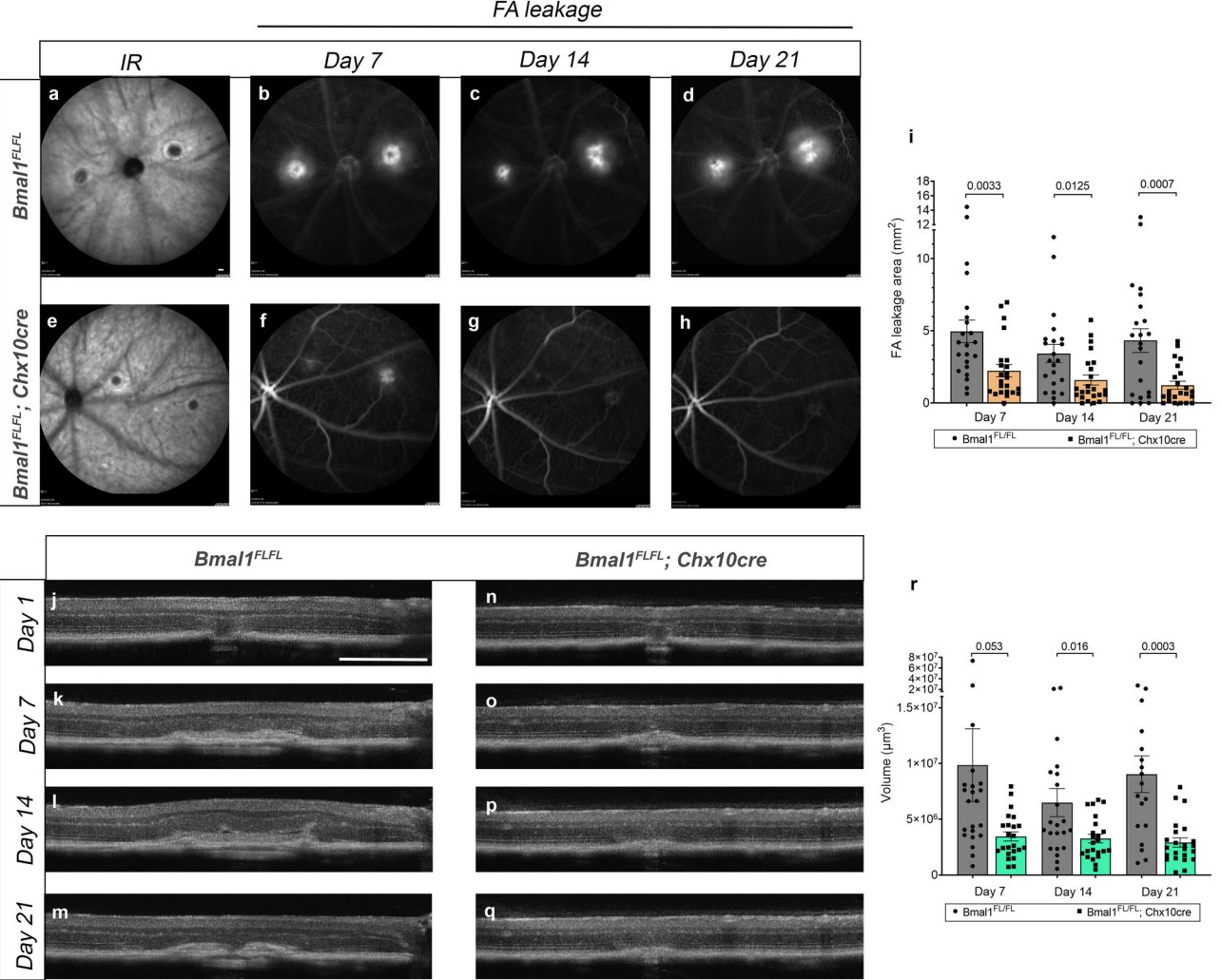

**Fig. 4 Dysregulated Bmal1 promotes neovascularization during CNV.** Adult mice were subjected to laser burns and evaluated by SLO on days 7, 14, and 21. Animals were injected with NaF (FA) and imaged to determine leakage area as demarcated by FA and SLO angiography. **a–h** Representative SLO images of Bmal1$^{FL/FL}$ at day 7, 14, and 21 post-laser injury from Bmal1$^{FL/FL}$ (**a, b, c, d**) controls and Bmal1$^{FL/FL}$; Chx10Cre mutant animals (**e, f, g, h**). **i** Quantitation of FA leakage showing significantly less leakage in the Bmal1$^{FL/FL}$; Chx10Cre animals. **j–q** Representative OCT-CNV volume scans of controls (**j, k, l, m**) and (**n, o, p, q**) mutant animals on day 7, 14, and 21 post injury. **r** Quantitation shows the CNV lesion volume is significantly reduced in Bmal1$^{FL/FL}$; Chx10Cre mutant animals (**r**). Error bars represented as SEM. Two-tailed $t$-test was performed to assess statistical significance. $n = 18$–24 for each experimental time point. Scale bar = 200 μm (**a**) & 0.4 mm (**j**).

Bmal1 highly regulates the semaphorin pathway. Semaphorin signaling is associated with developmental disorders of the nervous system and several retinopathies[14]. From the chip analysis, we identified Sema3A, 3D and 6D as the targets of Bmal1. We then performed in silico analysis of all three semaphorin promoter regions to establish whether any known canonical (5′-CANNTG-3′) BMAL1-binding E-box elements were present. Based on our in silico analysis analysis, Bmal1-binding sites are present on all three of these semaphorins. Interestingly, in the Sema6D and Sema3A promoter, there is a clock-binding E-box (CATATG) site close to the Bmal1 E-box. This region has considerable sequence homology in the mouse and human, to regulatory sequences of genes that are under circadian regulation (Fig. 6a).

Next we investigated whether Bmal1 can regulate the expression of these semaphorins. At P3, Sema3A and 3D transcripts levels did not differ between the Bmal1$^{FL/FL}$; Chx10Cre retina and the controls (Fig. 6e). However, Sema6D levels were significantly lower in the Bmal1$^{FL/FL}$; Chx10Cre animals (Fig. 6c). Next, we investigated whether the Sema6D

protein levels differed in the Bmal1$^{FL/FL}$; Chx10Cre retina. Surprisingly, though the total levels of the protein was not different, but the membrane bound receptor level was reduced in the Bmal1$^{FL/FL}$; Chx10Cre retina (Fig. 6b, d and Supplementary Figs. 5, 6). This data raises the possibility that Bmal1 may be regulating the targeting of this receptor on the membrane and thus affecting angiogenesis.

## Discussion
Our data strongly supports a role for a neuronal clock in regulating retinal angiogenesis by controlling the balance between the angiogenic and anti-angiogenic modulators. We have demonstrated that changing the light environment results in a hypertrophic retinal vasculature. These results indicate that the embryonic retina requires these light cues to establish a robust light driven circadian oscillator within the retina. There is increasing evidence to suggest that spectral composition of the light can have a huge impact on the circadian system[24–26]. In rodents, exposure to blue light can interfere with the clock gene expression in the SCN[27] while in humans blue light exposure in

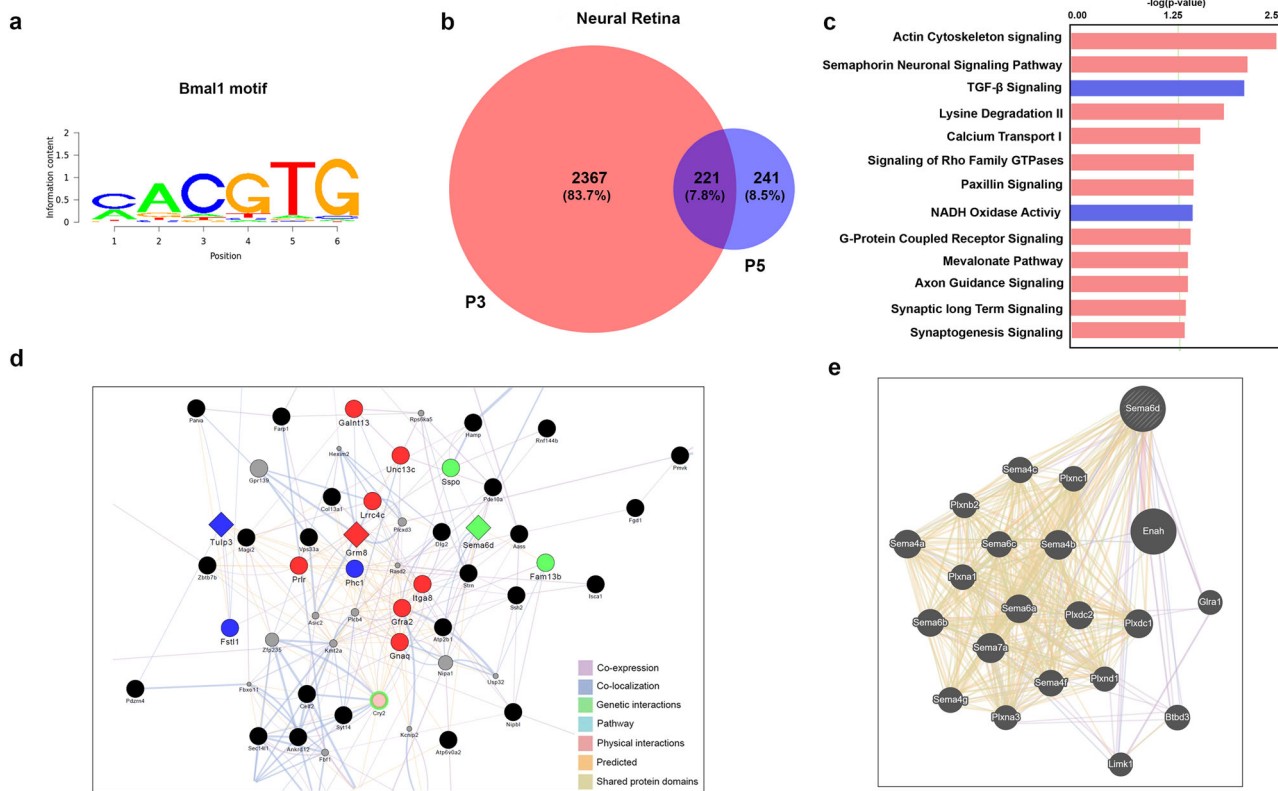

**Fig. 5 Bmal1 regulated pathways in the neural retina. a** Bmal1 (Arntl) motif used to validate Bmal1-binding sites from the large chip-seq dataset. **b** Venn diagram indicating Bmal1 target genes at P3 and P5. **c** GO terms for Bmal1 targeted pathways uniquely enriched at P3 (red bars) and P5 (blue bars) after the validation. **d** CytoScape and GeneMANIA interaction network of validated genes at P3 with co-expression and co-localization indicated by similar colors. Diamond shape denotes possible upstream gene of respective gene color network. Dim gray nodes represent query gene lists. **e** Network shows possible physical interactions, co-expression and genetic interactions with Sema6D gene.

the evening can suppress melatonin and negatively impact the wake sleep cycle[28]. Recent studies have shown that even very dim UV light can elicit neuronal responses within the SCN, thus further supporting a role for short wavelength of light in circadian functions[29]. It remains to be determined whether the embryonic retina is also sensitive to certain wavelengths of light. The genetic deletion of Bmal1, the core clock gene, results in a similar phenotype, suggesting that the light-mediated signals are regulating the expression of Bmal1. Bmal1 is unique among the core clock genes in that its disruption alone leads to arhythmicity[30]. Animals with global deletion of Bmal1, display various physiological deficits, including defective glucose/lipid metabolism, progressive arthropathy, early aging, decreased longevity, and infertility[31,32]. Bmal1 function within the retina is required for circadian rhythms of inner retinal visual processing, specifically for the circadian rhythms of the light-adapted ERG b-wave amplitude and b-wave implicit time[33]. Additional studies have shown that retinal loss of bmal1 has multiple age-related effects on both rod and cone pathways mediated by different mechanisms. The absence of Bmal1 also affects visual acuity and contrast sensitivity[34]. We have previously shown that loss of retinal Bmal1 controls the retinal progenitor proliferation by regulating the cell cycle. Furthermore, Bmal1 is also required for the spectral identity and function of the cone photoreceptors in the retina. In addition to Bmal1, both the transcripts and the protein levels for Per, Cry, Rora and Clock also show circadian variations in the adult retina[35–37]. Protein products of these clock genes regulate cellular processes during eye development. For example, deletion of Per1 and Per2 results in a number of developmental defects in the retina. Similarly, Cry1/Cry2 mutants exhibit defects in cone

photoreceptor function. These pleiotropic effects of loss of circadian proteins illustrates the complex role of circadian proteins in tissue development as well as function. Increasing evidence suggest that circadian proteins like Bmal1 play an essential role in organ physiology and tissue homeostasis that are independent of their circadian function[38]. However, many non-circadian functions of these proteins are still poorly understood. Though, it is conceivable that the effects on the retinal vasculature are mediated through a non-circadian function of Bmal1, it is highly unlikely given that we also see an effect in the Per2$^{FL/FL}$; Chx10Cre animals. Our results indicate that the embryonic retina requires these light cues to establish a robust circadian oscillator within the retina.

**Heterogeneity between the superficial, intermediate, and deep-layer vasculature.** The loss of Bmal1 and Per2 from the retinal neurons does not have a homogenous effect on the retinal vasculature. During early development, the superficial layer density is altered by the loss of Bmal1 and Per2. The superficial layer develops in close contact with the ganglion cell layer and the astrocytes, both of which can influence the growth of the vessels. Given that we have previously shown that melanopsin expressing intrinsically photosensitive retinal ganglion cells (ipRGCs) are required for normal growth of the superficial layer, it remains a possibility that the clock function is required in the ipRGCS. However, recent studies have shown that the opsin-5 expressing ganglion cells can also regulate retinal angiogenesis and hence the location of the clock could be in either or both of these ganglion cells. Interestingly, in the Per2$^{FL/FL}$; Chx10Cre animals, there is an opposite effect on the retinal vasculature. At P7.5, vessel

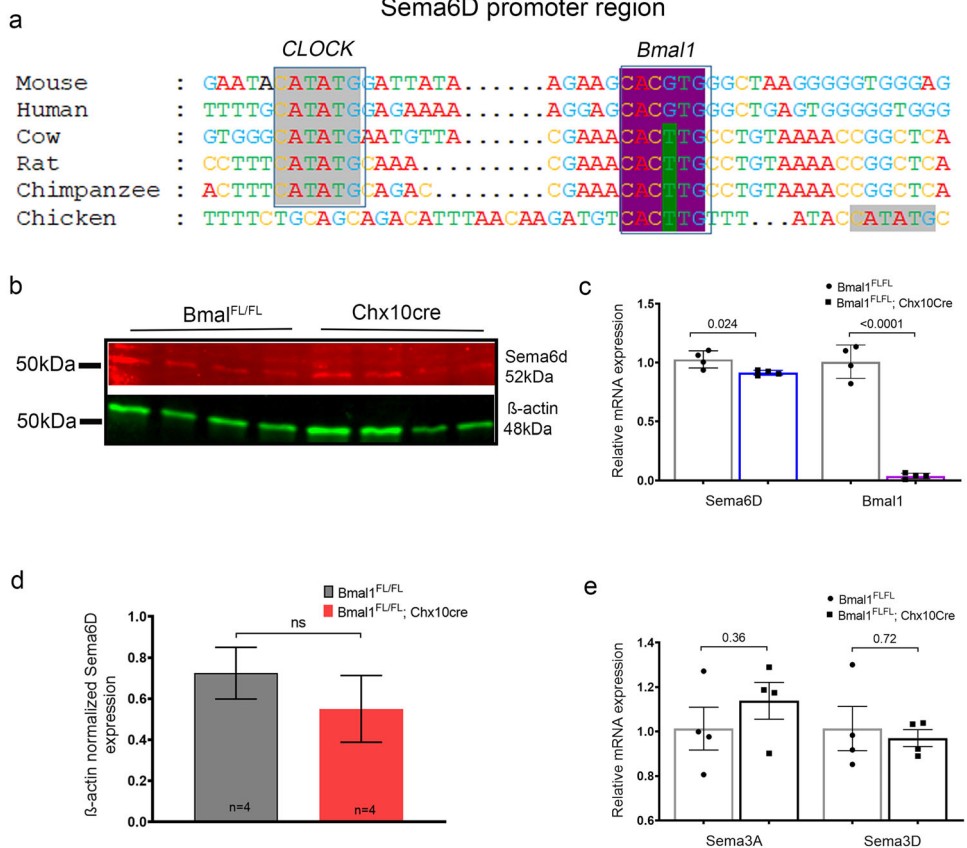

**Fig. 6 Bmal1 regulates Sema6D expression. a** Alignment of the region surrounding the conserved E-box-binding sequence for Bmal1 (highlighted in purple) and CLOCK (gray) in the promoter region of Sema6D, ~800 bp upstream of the transcription start site. **c** Quantitative RT PCR analysis for Sema6D transcript shows lower amount of transcript at P2.5 in the Bmal1FL/FL; Chx10Cre mutants compared to control Bmal1FL/FL animals. **b, d** Protein blots of retinal lysates probed with Sema6D antibody shows a reduction in the Sema6D protein levels in the Bmal1FL/FL; Chx10Cre retina. **e** Quantitative RT PCR analysis indicating that *Sema3A* and *Sema3D* mRNA levels are not different at P2.5 in the Bmal1FL/FL; Chx10Cre mutants compared to control Bmal1FL/FL animals.

density in the Per2FL/FL; Chx10Cre animals is less than the wildtype, but by P20.5, the superficial layer is similar between the WT and Per2FL/FL; Chx10Cre. This data suggests that the loss of Per2 from the retinal neurons results in accelerated growth and remodeling of the retinal vasculature. At P20.5, the deeper layer vessel density is similarly reduced in the Per2FL/FL; Chx10Cre and remains to be determined whether these changes in the deeper plexus are transient. Conversely, in the Bmal1FL/FL; Chx10Cre animals, the hyperproliferative phenotype detected at P7.5 remains persistent but the deeper layer plexus is unaffected. These regional effects on the vessel density could be explained by differences in the neuronal population that are driving the growth of the vessels. For example, deletion of Vegfa from the amacrine and horizontal cells, despite causing a substantial reduction in Vegfa transcript levels in the INL, does not affect either the superficial or deep layer plexus. However, it does result in severe attenuation of the intermediate plexus. Alternatively, Vegfa overexpression in amacrine and horizontal cells induces massive neovascularization both in the INL and IPL[17]. Our observations are similar and suggest that clock genes within these neuronal population are differentially regulating the factors that drive angiogenesis. Surprisingly, there was no difference in the levels of Vegfa transcript in Bmal1FL/FL; Chx10Cre retina at P3 (Supplementary Fig. 4), suggesting that neuronal clock is regulating other angiogenic factors. This finding is opposite to what has been reported in zebrafish, where Bmal1 regulates developmental angiogenesis by directly binding to the Vegfa promoter[39]. The difference could be

due to species differences, cell type specific function of Bmal1 or the timing of assessments. Nevertheless, our data supports a role for the neuronal circadian clock in the regulation of retinal angiogenesis.

**Semaphorin in angiogenesis and OIR.** At P3, semaphorin signaling pathway is the most dominant pathway influenced by Bmal1. Semaphorins are a large and diverse family of widely expressed secreted and membrane-associated proteins that play an essential role in neuronal development, angiogenesis, immunoregulation, and cancer. The best-characterized receptors for mediating semaphorin signaling are membranes of the neuropilin and plexin families, but the molecular mechanisms of semaphorin signaling are still poorly understood. Our data suggest that Sema3A, 3D, and 6D expression is highly regulated by neuronal Bmal1. The function of these semaphorins is largely confined to the growth of the axons[40–42], except Sema3A, which is also involved in pathological angiogenesis. In an OIR model, deletion of Sema3A in retinal ganglion cells results in normal vascular regeneration in the ischemic retina. Thus, it appears that Sema3A is secreted by hypoxic neurons and inhibits the vascular regeneration of the retina while enhancing pathologic retinal neovascularization[43]. It is not known whether Sema3D or 6D can function similarly in the retina but all three of these have functional role in either inhibiting or promoting tumor angiogenesis[44,45]. Neither Sema3A nor 3D transcripts are altered in the Bmal1FL/FL; Chx10Cre retina but the Sema6D transcript is reduced at P3 (Fig. 6e). It is conceivable that

Bmal1 does not regulate the class 3 semaphorins at P3 and these semaphorins do not play any functional role in retinal angiogenesis. It will be interesting to determine whether dysregulated expression of these semaphorins occurs under stress and injury due to a dysfunctional circadian clock. Bmal1 occupancy of the semaphorin promoter is extremely dynamic, as none of these semaphorins were bound by Bmal1 at P5 (Supplementary Table 3). Thus, our data indicates that Bmal1 is transiently regulating semaphorin expression in the neurons perhaps to prevent aberrant growth of the blood vessel.

**Circadian clock in regulation of angiogenesis or retinopathies**. Our analysis suggests a role for Bmal1 signaling in driving neovascularization. Accordingly, in animals with oxygen-induced retinopathy, loss of Bmal1 causes a drastic reduction in neovascularization. Moreover, the regrowth of the vessels into the hypoxic area is accelerated. In these animals the retinopathy is induced during development and thus it can be argued that Bmal1 is important during development. However, we observed a similar reduction in choroidal neovascularization in the laser-induced model with accelerated recovery of the vasculature. Thus, we propose that there is a common mechanism by which injury leads to a local disruption of the circadian clock, which is in turn results in local upregulation of angiogenic factors. Under these conditions, loss of Bmal1 could be protective as it will result in inactivation of the dysregulated clock that promotes the progression of retinal pathologies. A similar observation was noted in an animal model of diabetic retinopathy, where loss of Bmal1 ameliorates retinal pathologies associated with diabetic retinopathy like the thinning of the retina and upregulation of molecular markers for DR[46]. Conversely, animals with global loss of Per2, exhibit vascular phenotypes that are similar to what is observed in diabetic animals. The authors only examined the retina but the animals were not subjected to any injury to assess for the effects of Per2 on neovascularization[47,48]. In our analysis, loss of Per2 from the retina has a similar effect on neovascularization, suggesting that inactivation of the dysregulated clock can be protective against ischemia induced neovascularization.

In conclusion, the mammalian circadian clock is emerging as a critical component for several disease processes that include aging-related disorders, retinopathies, neurodegenerative diseases, and cancer[49–53]. Angiogenesis is one of the most important processes for the progression of these diseases[23,54,55]. Therefore, a tremendous effort has been directed towards anti-angiogenic therapy in cancer, retinopathies, metabolic diseases or chronic inflammatory diseases and proangiogenic therapy in neurodegenerative disorders, myocardial infarction, stroke or diabetic peripheral vascular disorders. The circadian clock may affect angiogenesis by directly regulating both pro and anti-angiogenic factors. Interestingly, clock malfunction might contribute to the pathogenesis of the disease but loss of the clock genes themselves may protect from aberrant and leaky vessel growth. As our knowledge continues to expand in regards to the significant role of circadian clock genes in the pathophysiology of the disease states, future targeting of the underlying pathways that control mammalian circadian rhythm may hold the key for the development of novel therapies against aging-related disorders, retinopathies, neurodegenerative disease, and tumorigenesis.

## Methods

**Mouse strains**. Transgenic mice carrying Bmal1$^{FL/FL}$ (stock no. 018985) tdTomato (stock no. 007914), Chx10Cre (stock no. 005105) were purchased from The Jackson Laboratory (Bar Harbor, ME). Bmal1$^{FL/FL}$; Chx10Cre and Per2$^{FL/FL}$; *Chx10Cre* were generated as previously described[9,10]. Inducible Pdgf-icreER were purchased from MRC[56]. For endothelial cell gene deletion canola oil-dissolved tamoxifen (T6648-1G, Sigma St. Louis, MO, USA) was injected intraperitoneally at 15 μg/g body weight from day of birth (P0.5) till P2.5. Pdgf-icreER were crossed to Bmal1$^{FL/FL}$ to generate

Bmal1$^{FL/FL}$; Pdgf-icreER. Similar scheme was used to generate Bmal1$^{FL/FL}$; tdTomato$^{FL/+}$; Pdgf-icreER (mutant) or tdTomato$^{FL/+}$; Pdgf-icreER (control). All WT animals used in the study are derived from C57BL/6J and the transgenic mice are also from the C57BL/6 J background. All animals were maintained in a light/dark (12 L:12D) and fed ad libitum with normal chow. Embryonic ages were determined based on the detection of the vaginal plugs in the pregnant females and marked as E1 (embryonic day 1). At E13 cages were assigned to DD conditions and maintained till E16. LL cages were assigned to LL conditions at E15 and maintained till postnatal day P4.5. LD cages were maintained in 12 h light/dark cycle, DD cages were kept in constant darkness and LL cages were maintained in constant light throughout the experimental period. The LL lighting conditions were the same as the regular mouse room light with standard fluorescent lighting (0.51 KLx). An infrared night vision goggle was used to monitor the animals in the DD room. Animal use protocol was approved by the Institutional Animal Care and Use Committees at Cleveland Clinic (IACUC), and all procedures followed the Association for Research in Vision and Ophthalmology (ARVO) guidelines for the use of animals in vision research.

**Quantitative PCR**. Total RNA was extracted from neural retina using an RNeasy Mini Kit (Qiagen, Hilden, Germany) according to the manufacturer's instructions. 250 ng of DNASe-treated RNA was reverse-transcribed using a Versoenzyme CDNA synthesis kit (Thermo Fisher Scientific, Waltham, MA, USA). cDNA amplification was performed using gene-specific primer sequences are mentioned in Supplementary Table 1. Real-time PCR was performed using Quantstudio3 (Biorad) using a Radiant SYBR green Low-ROX qPCr mix (Alkali Scientific, Pompano Beach, FL, USA). Each assay was run in biological duplicate or triplicate. Expression of beta actin was used as the endogenous control for mRNA relative quantifications. Relative quantification of gene expression levels between samples was performed using the comparative Ct (threshold cycle number) method[57]. Subsequently, the 2$^{ΔΔCt}$ method was used to evaluate the relative expression level (fold change).

**ddPCR**. Reaction mixtures containing primers (250 nM), template (250 ng) and QX200™ ddPCR™ EvaGreen Supermix with 1x as final concentration (Bio-Rad: 1864034) of total volume 20 μl were used according to the manufacturer's instructions. Droplet generation and transfer of emulsified samples to PCR plates was performed according to manufacturer's instructions (Bio-Rad: QX200™ Droplet Generator). The ddPCR plate was sealed with a foil heat seal (Bio-Rad: 1814040) and the PX1™ PCR Plate Sealer (Bio-Rad: 181–4000). For transcript count ddPCR (Bio-Rad: QX200 Droplet Digital PCR (ddPCR™) System) was used. For ddPCR technology, the absolute quantity of DNA per sample (copies/μl) was processed using QuantaSoft (v.1.7.4.097) and for analysis QuantaSoft analysis Pro (1.0) program was used.

**Endothelial cell isolation by fluorescence-activated cell sorter (FACS)**. Post enucleation, retinas were dissected in ice cold DMEM from P7.5 pups of the following genotype: Bmal1$^{FL/FL}$; tdTomato$^{FL/+}$; Pdgf-icreER. Retinal dissociation was done in *Dulbecco's Modified Eagle Medium* (DMEM) containing 10 mg/ml of collagenase A (10103578001, Roche, USA) and 3U/ml of DNase1 (D5025, Sigma, St. Louis, MO, USA) at 37 ºC for 20 min with pipetting every 10 min. Cells were passed through a 70 μm cell strainer (Fisher Scientific, 22363548) and centrifuged for 8 min at 3000 rpm. After PBS washes cells were reconstituted in PBS containing 1 mM EDTA. TO-PRO (T3605, Themo Scientific, Waltham, MA, USA) or DAPI were used for identification of dead cells. Both GFP and tomato double-positive population were sorted and collected in RLT buffer for RNA isolation using the RNeasy mini kit (Qiagen, Germantown, MD, USA).

**Western blot**. Isolated retinas were lysed in lysis buffer for protein extraction as previously described[19]. Isolation of membrane and cytoplasmic fraction was performed as described[58]. Blots were probed with antibodies against Bmal1 (ab3350; Abcam, Cambridge, MA, USA), Sema6D (AV49583, Sigma), and β-actin (4970 L; Cell Signaling Technology, Danvers, MA, USA), β-tubulin (9F3; Cell Signaling Technology, Danvers, MA, USA). Immunoblots were visualized using IRDye 680RD donkey anti-mouse/rabbit antibody or 800CW donkey anti-mouse/rabbit secondary antibody (925-32213; Li-Cor Biosciences, Lincoln, NE, USA). Membranes were scanned with an Odyssey infrared scanner (Li-Cor Biosciences).

**Whole-mount immunofluorescence of retina**. Retinas were dissected and fixed in 4% PFA for 1–2 h at room temperature (RT, washed three times in PBS for 10 min, permeabilized for 1 h using 1% Triton X-100, blocked in 0.03% Triton X-100 with 3% BSA for 1 h at RT. Retinas were incubated with Isolectin Alexa 488-Isolectin IB4 (1:200) (121411; Life technologies, Carlsbad, CA) for 24–48 h at 4 ºC. Retinal flat mounts were mounted in mounting medium. Imaging was done using a Leica laser scanning confocal microscope[59]. Whole retinal flat-mount or cross-sectional images were combined images that were created from stitching individual 10x or 20x tile images.

**Cosinor analysis**. The rhythmicity of gene expression was evaluated using web-based tool (https://cosinor.online/app/cosinor.php) and further confirmed with cosinor analysis, as defined by the equation [$Y$ = mesor + (amplitude* − cos(2*$p$*

($X$ − acrophase)/wavelength][60], with a constant wavelength of 24 h, and cosine curve as the alternative hypothesis. Once rhythmicity was established, the cosine curve parameters were calculated: mesor (the mid-value of the cosine curve representing a rhythm-adjusted mean), amplitude (the difference between the peak and the mean value of the cosine curve), and acrophase (the time of the peak of fitted curve, each individual parameters were compared. Two-way ANOVA was used with a multiple comparison by the Sidak–Bonferroni method to compare the control LD and LL-exposed groups. Data are reported as the mean ± SEM of 4–5 animals. Two-tailed $t$-tests were performed to compare the trends in amplitude, mesor and acrophase. All tests were performed in Graph Pad Prism 6 & 9 (GraphPad Software Inc., San Diego, CA, USA).

**OIR.** The ROP model used in this study is based on the protocol established by Smith et al.[21]. In brief, nursing mothers and their pups are placed in 75% oxygen (hyperoxia) in the Biospheric animal hyperoxia system (Biospherix Ltd.) for 5 days (P7.5 through P12.5, date of birth is P0.5). Animals are returned to the normal oxygen conditions (normoxia) at P12.5. Oxygen-induced vascular pathologies like vaso-obliteration and neovascularization is automatically quantified using deep-learning segmentation software available for free at an open source repository[61].

**Chromatin immunoprecipitation (ChIP) sequencing.** A detailed protocol for the ChIP analysis is reported previously with the exception that retina was isolated from P3 and P5 and the time of the tissue harvest is ZT = 4 for P3 and P5. In brief, ChIP assay was performed using the rabbit anti-Bmal1 antibody (2 μg/sample, 12-370, Abcam, Millipore, Billerica, MA, USA). An Ultrasonicator (Model S2, Covaris, Inc.Woburn, MA, USA) was used for fragmentation of chromatin DNA. All other protocols were followed as described[9] and the analysis was performed by the Genomics Facility-University of Chicago, Chicago, IL, USA. Sequencing was done using HiSeq 4000 (Illumina, San Diego, CA, USA). Motif-specific analysis was done using Cistrome and Memechip tools and the resulting analysis was further confirmed with https://epd.epfl.ch//index.php.

**Optical coherence tomography.** Ultra-high-resolution spectral domain optical coherence tomography (OCT) system (Envisu R2210 UHR Leica Microsystems Inc.) was used for in vivo cross-sectional imaging. The scan parameters used were 1.8 × 1.8 mm rectangular volume scan, 1000 a-scans/200 b-scans averaged three times per b-scan. The B-scan was used to measure the width ($a$) and depth ($b$) of the lesion. The volume intensity projection (VIP) image was used to measure the length ($c$) of the lesion. The lesion volume was calculated using the formula $A = 43πabc$. Immediately after the laser injury, each animal was scanned for ruptured Bruch's membrane to confirm successful CNV induction. The inclusion criteria to include a lesion in the quantitative analysis is as described[62]. Each CNV lesion was then followed over time using the same scan parameters at day 7, 14, and 21. The volume of each lesion was calculated with OCT volumetric scans using previously described methods[62].

**Scanning laser ophthalmoscopy.** Animals were imaged with cSLO (Heidelberg SPECTRALIS, Heidelberg Engineering, Germany) as previously described[63]. Briefly, for all post laser imaging, fluorescein angiography (FA) was performed at the RPE-choroid interface and vitreous–retina interface for early-phase and late-phase CNV leakage analysis. For intraperitoneally injected Na Fluorescein (1% in saline, 50 μl) (AL-FLUOR 10%, Akron Inc.) early-phase images of fluorescein angiography (FA) were taken at ~3 min and late-phase images were taken at ~6 min. CNV images were analyzed in ImageJ (Version 1.52a) by measuring area of leakage.

**Statistical analysis.** Statistical significance was determined using Graphpad Prism 9.0[64]. Results comparing control to mutant groups were analyzed using Student's two-tailed $t$-test. The LD and DD comparison were done using two-way repeated measures ANOVA followed by pairwise multiple comparison using Tukey's test or Sidak–Bonferroni method. All data was assessed for normalized distribution.

**Reporting summary.** Further information on research design is available in the Nature Research Reporting Summary linked to this article.

## Data availability

The authors declare that data supporting the findings of this study are available within the article, and its supplementary information files. Raw data and related files for CHIP-seq is deposited to GEO and the accession number is GSE207271.

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

## Acknowledgements

The authors are grateful to the Biological Resources Unit and Veterinary Services at Cleveland Clinic for their assistance with mouse colony management. The authors are thankful to Dr. Jonathan Sears and Dr. George Hoppe for their assistance with the OIR experiments, Dr. Amy Nowacki for her assistance with statistical analysis. The authors are thankful to Lerner Research Institute Flow Cytometry for the assistance with cell sorting. The authors thank Ricky Chan for assistance with CHIP-seq data analysis. Supported by grants from the U.S. National Institutes of Health/National Eye Institute EY027077-01 (S.R), RPB1503 (S.R.), National Eye Institute P30-EY025585 Core Grant and Research to Prevent Blindness Challenge Grant.

## Author contributions

This study was conceived by S.R. Experimental design was done by S.R., V.K.J. Acquisition of data was done by S.R., V.K.J., O.B.S., R.D.F., K.W., R.S. R.A.L. provided essential tools. Analysis and interpretation of data was done by S.R., V.K.J. Drafting of the paper was done by S.R., V.K.J.

## Competing interests

The authors declare no competing interests.
