## [Peer Review File · Communications Biology]

Reviewers' comments:

Reviewer #1 (Remarks to the Author):

The main problem with manuscript can be summarized as follow:

1) the authors do not mention what has been previously reported about the role of Bmal1 in the retina. A few studies have investigated this aspect and the authors should add a paragraph summarizing what is know.

2) the authors do not mention previous work that has investigated clock gene expression during development in the retina (e.g., Expression and light sensitivity of clock genes Per1 and Per2 and immediate-early gene c-fos within the retina of early postnatal Wistar rats.

Matejů K, Sumová A, Bendová Z.J Comp Neurol. 2010 Sep 1;518(17):3630-44. doi: 10.1002/cne.22421.PMID: 20589906)

3) the authors should also summarize what is know about circadian gene expression in the mouse retina.

4) the authors lack the understanding of circadian term. If a gene is no rhythmic in DD there is no circadian pattern. The terminology used in the manuscript about circadian and daily is confusing and most of the time wrong.

Minor points:

the English needs some improvement (e.g., data is plural) and the name of the gene and protein need to consistent with the nomenclature

The statistical analysis need to provide more info about: 1) can parametric statistic be use with the data collected (e.h., equal variance, etc.) and does the stat used has enough power

Reviewer #2 (Remarks to the Author):

This is an interesting and well-written manuscript. I do have several comments that should be addressed:

1. I am not quite clear why the authors have chosen an ROP model. It seems from the discussion that the authors are mainly interested in DR. Why wasn't a model chosen that more closely resembles human DR?

2. There is increasing evidence that the spectral components of the light play an important role in clock regulation. The authors need to provide more details and discuss this topic.

3. The introduction is too long and shall be shortened by at least 50%.

Reviewer #3 (Remarks to the Author):

This manuscript by Jidigam et al. is an attempt to establish causality between circadian clock malfunction in the retina and pathological neovascularization. The authors show that manipulating environmental light early in development leads to defects in clock gene expression (and presumably clock dysfunction) and angiogenic defects in the retina. Elimination of clock core genes (Bmal1, Per2) in the retinal neurons attenuates pathological neovascularization. Furthermore, the data indicate that Bmal1 may control vascularization through semaphorin signaling. The main conclusion of the paper is that neovascularization in the retina, a hallmark of some retinopathies, may result from clock dysregulation. Eliminating/blocking clock function could represent a therapeutic strategy to block pathological neovascularization.

I enjoyed reading this paper. The rationale is clear and the approach sound. A large amount of data is presented and globally the data support the conclusions. I have a few minor comments/clarifications. There are also some strong statements that will have to be soften. :

- Abstract: clarify what is "disruptive light cycle"

- "Deregulation": I believe "dysregulation" might be more correct (throughout the text).
- Abstract: last sentence, consider "therapeutic silencing of the retinal clock"
- Line 74: add recent reviews on retinal clocks (i.e. Felder-Schmittbuhl et al., 2018).
- Lines 77-79: "However, it is not CLEAR..." and cite Bery et al., 2022, Bagchi et al., 2020. Some impacts of clock function on retinal development have been described in Sawant et al., 2017 + 2019.
- Line 92: +refs
- Lines 94-96: "However..." please clarify.
- Lines 254-255: I don't follow that sentence, please clarify
- Line 257: $p=0.056$ is NOT significant
- Line 264 (and many other places in the manuscript): please clearly state when and for how long the animals (pregnant dams) have been in LL or DD.
- Lines 266-267: Bmal1 is essential: but Per2 is still rhythmic!
- Line 270: this reference is about melanopsin, which brings the following question: where is the clock that influences the growth of retinal vasculature? in ipRGCs? in OPN5+ RGCs? There should be some discussion about the possible location of the clock.
- Line 286: cre expression follows a mosaic pattern in CHX10cre retinas.
- Line 295: there should be a few sentences to introduce the reader to BMAL1-positive loop vs PER2-negative loop, and that removal of either of the two would be expected to lead to opposite outcomes. It is buried in the text but should be very clear.
- Is there a phenotype in Bmal1^{ff} (or Per2^{ff}) vs WT (B6)?
- Lines 302-308: add references
- Lines 320-328: please remind the reader of the timing/age of the injections.
- Line 336: "To test", not "To demonstrate"
- Line 337: adult: age?
- Lines 372-374: add references
- Lines 388-389: this hypothesis could be easily tested: day vs night, Bmal1^{+/+} vs Bmal1^{-/-}
- Line 392: I don't like the "light-driven", in the absence of light, the clock may normally develop with a delay, you just don't know. In addition, you measured only 1 time point. Consider revising lines 406-408 and the abstract as well.
- Lines 424-426: add references.
- Lines 445-446: +refs.
- Line 460: "suggests"
- Line 485: "disruption" is not correct. Bmal1^{ko} also have a disrupted clock. Consider "clock malfunction".
- Line 340: "at day 4, 14, and 21" post-injury?
- Figure1: panels E-H are redundant.

Dear Reviewers,

Thank you for your positive critiques and suggestions. We have incorporated these suggestions into the manuscript shown as highlighted texts. These suggestions have greatly improved the manuscript and we hope that we have satisfactorily addressed all the reviewer's comments

Reviewers' comments:

Reviewer #1

RI: The main problem with manuscript can be summarized as follow:

1) the authors do not mention what has been previously reported about the role of Bmal1 in the retina. A few studies have investigated this aspect and the authors should add a paragraph summarizing what is know.

AR: We apologize for the oversight and have now included this information in the discussion, (Line 372-385).

RI: the authors do not mention previous work that has investigated clock gene expression during development in the retina (e.g., Expression and light sensitivity of clock genes Per1 and Per2 and immediate-early gene c-fos within the retina of early postnatal Wistar rats.

Matejů K, Sumová A, Bendová Z. J Comp Neurol. 2010 Sep 1;518(17):3630-44. doi: 10.1002/cne.22421. PMID: 20589906)

AR: As the reviewer has suggested we have included this reference and additional information regarding clock gene expression in the developing retina (Introduction line 68 -70).

RI: the authors should also summarize what is know about circadian gene expression in the mouse retina.

AR: See response above

RI: the authors lack the understanding of circadian term. If a gene is no rhythmic in DD there is no circadian pattern. The terminology used in the manuscript about circadian and daily is confusing and most of the time wrong.

AR: We apologize for our lack of clarity. We are aware that rhythmicity under constant conditions is what defines the circadian clock. However, we think that during development the circadian clock machinery is not yet established and hence we don't expect to see rhythmicity and light entrainment in the embryonic retina. We believe that the developing retina is using light signals to establish circadian clocks within different cells. Thus the system is sensitive to light input and as reviewer 3 points out, in the absence of light there is perhaps a delay in the setting up of the retinal clock.

We have also clarified the terminology in the manuscript as the reviewer has pointed out.

Minor points:

R1: the English needs some improvement (e.g., data is plural) and the name of the gene and protein need to be consistent with the nomenclature

AR: We apologize for the editorial errors and have edited the manuscript

R1: The statistical analysis needs to provide more info about: 1) can parametric statistics be used with the data collected (e.g., equal variance, etc.) and does the test used have enough power

AR: Thank you to the reviewer for bringing this to our attention. We assumed the distribution was normal in the wildtype and hence we used a parametric test. As the reviewer alludes, this test is also more stringent compared to the non-parametric test and in some comparisons we did not have enough sample size to assume a normal distribution. After the reviewer's suggestion, we reanalyzed all our data for normality and our data shows a normal distribution. In addition, the P-values between the parametric versus non-parametric tests did not change. We also consulted a biostatistician Dr. Nowacki to help us with the statistical analysis and the analysis is still valid. We have now included more explanation for the test in the material and methods.

Reviewer #2 (Remarks to the Author):

This is an interesting and well-written manuscript. I do have several comments that should be addressed:

R2: I am not quite clear why the authors have chosen an ROP model. It seems from the discussion that the authors are mainly interested in DR. Why wasn't a model chosen that more closely resembles human DR?

AR: Our goal for this manuscript was to determine the role of clock genes during development. ROP was chosen because it is a model for developmental pathogenesis. Since neovascularization was drastically reduced in the OIR animals, we decided to investigate whether this is true for all types of neovascularization and hence we used the laser induced CNV model. In both the models, loss of *Bmal1* causes drastic reduction in neovascularization. This is exciting data for us and we intend to test the clock function in the DR models in the future.

R2: There is increasing evidence that the spectral components of the light play an important role in clock regulation. The authors need to provide more details and discuss this topic.

AR: We agree with the reviewer that the spectral components of the light are important in the entrainment of the circadian clock. We did not include this information because the light experiments were done with a standard mouse room lighting (white light) and we have not addressed which wavelength of light is mediating these effects. It would be interesting to investigate whether the embryonic retina is more responsive to certain wavelengths of light. However, for this manuscript, we have limited our analysis to a full spectrum light source to focus on the clock genes in the retina. As per the reviewer's suggestion, we have added this point in the discussion (line 360-367) and we have also clarified in the methods section the LL conditions.

R2: The introduction is too long and shall be shortened by at least 50%.

AR: We have now shortened the introduction and hope that we have satisfied the reviewer.

Reviewer #3 (Remarks to the Author):

This manuscript by Jidigam et al. is an attempt to establish causality between circadian clock malfunction in the retina and pathological neovascularization. The authors show that manipulating environmental light early in development leads to defects in clock gene expression (and presumably clock dysfunction) and angiogenic defects in the retina. Elimination of clock core genes (Bmal1, Per2) in the retinal neurons attenuates pathological neovascularization. Furthermore, the data indicate that Bmal1 may control vascularization through semaphorin signaling. The main conclusion of the paper is that neovascularization in the retina, a hallmark of some retinopathies, may result from clock dysregulation. Eliminating/blocking clock function could represent a therapeutic strategy to block pathological neovascularization. I enjoyed reading this paper. The rationale is clear and the approach sound. A large amount of data is presented and globally the data support the conclusions. I have a few minor comments/clarifications. There are also some strong statements that will have to be softened. :

R3: Abstract: clarify what is “disruptive light cycle”

AR: We apologize for the error and have changed it to “constant light conditions”.

R3: “Deregulation”: I believe “dysregulation” might be more correct (throughout the text).

AR: We have now changed the deregulation to dysregulation in the entire manuscript.

R3: Abstract: last sentence, consider “therapeutic silencing of the retinal clock”

AR: Thank you for the suggestion. We have changed therapeutic targeting of the circadian clock to “therapeutic silencing of the retinal clock”.

R3: Line 74: add recent reviews on retinal clocks (i.e. Felder-Schmittbuhl et al., 2018).

AR: We have included some recent reviews Felder-Schmittbuhl et al., 2018, Besharse et al., 2016, DeVera et al., 2019 (line 66).

R3: Lines 77-79: “However, it is not CLEAR...” and cite Bery et al., 2022, Bagchi et al., 2020.

Some impacts of clock function on retinal development have been described in Sawant et al., 2017 + 2019.

AR: We have corrected the statement with the updated references (line 68-70).

R3: - Line 92: +refs

AR: We have now changed this statement.

R3: Lines 94-96: “However...” please clarify.

AR: We have now changed the sentence in the line 77-79.

R3: Lines 254-255: I don't follow that sentence, please clarify

AR: We have deleted that statement.

R3: Line 257: $p=0.056$ is NOT significant

AR: We apologize for our error, we have changed the sentence to not significant (line 221).

R3: Line 264 (and many other places in the manuscript): please clearly state when and for how long the animals (pregnant dams) have been in LL or DD.

AR: We would like to thank the reviewer for bringing this to our attention. We have now added the additional information in the methods section (line 100 - 105) and in the results (line 227-228).

R3: Line 270: this reference is about melanopsin, which brings the following question: where is the clock that influences the growth of retinal vasculature? in ipRGCs? in OPN5+ RGCs? There should be some discussion about the possible location of the clock.

AR: Thank you to the reviewer for bringing up this point. The location of the clock is not known and we hope that we can address this question in a future manuscript. We have included this point in the discussion line 397-402.

R3: Line 286: cre expression follows a mosaic pattern in CHX10cre retinas.

AR: We have edited the statement to clarify that the Cre expression in the RPC is not uniform line 253.

R3: Line 295: there should be a few sentences to introduce the reader to BMAL1-positive loop

vs PER2-negative loop, and that removal of either of the two would be expected to lead to opposite outcomes. It is buried in the text but should be very clear.

AR: As the reviewer pointed we have added additional information in the introduction and the result section (line 248-251 & 267).

R3: Is there a phenotype in *Bmal1*^{ff} (or *Per2*^{ff}) vs WT (B6)?

AR: No, we have compared both the *Bmal1*^{F1/F1} as well as *Per2*^{F1/F1} to WT and we do not see any significant change in the vessel density. We have also compared vessel density between *Chx10*Cre transgene and wild type and do not see any difference.

R3: Lines 302-308: add references

AR: We would like to thank reviewer for pointing out. We have now added the references

Nian, S., Lo, A.C.Y., Mi, Y. *et al.* 2021 Neurovascular unit in diabetic retinopathy: pathophysiological roles and potential therapeutical targets (line-272)

&

Smith et al., 1994 Oxygen induced retinopathy in the mice (line- 276)

R3: Lines 320-328: please remind the reader of the timing/age of the injections.

AR: The injection were performed at Day of birth for a duration of 3 days. This information is included in the material and methods and also in the result section. (line 290-291) .

R3: Line 336: “To test”, not “To demonstrate”

AR: As reviewer pointed we have now has now changed the text accordingly.

R3: Line 337: adult: age?

AR: We thank reviewer for pointing out. The animals were 6 months old and we have included the age in the text (line 302).

R3: Lines 372-374: add references

AR: We have now added the below reference.

Cerani A, Tetreault N, Menard C, Lapalme E, Patel C, Sitaras N, Beaudoin F, Leboeuf D, De Guire V, Binet F, Dejda A, Rezende FA, Miloudi K, Sapieha P. Neuron-derived semaphorin 3A

is an early inducer of vascular permeability in diabetic retinopathy via neuropilin-1. *Cell Metab.* 2013 Oct 1;18(4):505-18. doi: 10.1016/j.cmet.2013.09.003.

R3: Lines 388-389: this hypothesis could be easily tested: day vs night, Bmal1^{+/+} vs Bmal1^{-/-}—

AR: The reviewer is correct, and we are working on testing this hypothesis for a future manuscript.

R3: Line 392: I don't like the "light-driven", in the absence of light, the clock may normally develop with a delay, you just don't know. In addition, you measured only 1 time point. Consider revising lines 406-408 and the abstract as well.

AR: We agree with the reviewer and have edited the statement in the abstract as well.

R3: Lines 424-426: add references.

AR: The following reference was added to the text (line 416). Usui Y, *et al.* Neurovascular crosstalk between interneurons and capillaries is required for vision. *The Journal of Clinical Investigation* **125**, 2335-2346 (2015).

R3: Lines 445-446: +refs.

AR: Reference was added in the line 437. Joyal J-S, *et al.* Ischemic neurons prevent vascular regeneration of neural tissue by secreting semaphorin 3A. *Blood* **117**, 6024-6035 (2011).

R3: Line 460: "suggests"

AR: As the reviewer pointed out we have now changed the sentence to "Our analysis suggests a role for Bmal1 signaling in driving neovascularization".

R3: Line 485: "disruption" is not correct. Bmal1^{ko} also have a disrupted clock. Consider "clock malfunction".

AR: We have now changed the sentence to "Interestingly, malfunction of the clock might contribute to the pathogenesis of the disease but loss of the clock genes themselves may protect from aberrant and leaky vessel growth".

R3: Line 340: "at day 4, 14, and 21" post-injury?

AR: As the reviewer pointed we have now added post injury (line 304 & 311).

R3: Figure1: panels E-H are redundant.

AR: As per the reviewer's request, we have removed the panels E-H in Figure 1.

REVIEWERS' COMMENTS:

Reviewer #1 (Remarks to the Author):

I am satisfied with the correction made in the revised version

Reviewer #3 (Remarks to the Author):

The authors have done a great job in revising the manuscript. In particular, they have improved the clarity of the statistical analysis. I have no further comments.